# Data Selection Through Iterative Self-Filtering for Vision-Language Settings

**Andrei Liviu Nicolicioiu**                                           *andrei.nicolicioiu@mila.quebec*
*Mila, Université de Montréal*

**Sarvjeet Singh Ghotra**                                   *sarvjeet-singh.ghotra@mila.quebec*
*Mila, Université de Montréal*

**Morgane M. Moss**                                                   *morgane.moss@mila.quebec*
*Mila, Université de Montréal*

**Aaron Courville**                                                       *courvila@mila.quebec*
*Mila, Université de Montréal*

**Reviewed on OpenReview:** *https://openreview.net/forum?id=FO9NfCXuCe*

## Abstract

The availability of large amounts of clean data is paramount to training neural networks. However, at large scales, manual oversight is impractical, resulting in sizeable datasets that can be very noisy. Attempts to mitigate this obstacle to producing performant vision-language models have so far involved heuristics, curated reference datasets, and using pre-trained models. Here, we propose a novel bootstrapped method in which a CLIP model is trained on an evolving, self-selected dataset. This evolving dataset constitutes a balance of filtered, highly probable clean samples, as well as diverse samples from the entire distribution. Our proposed Self-Filtering method iterates between training the model and selecting a subsequently improved data mixture. Training on vision-language datasets filtered by the proposed approach improves downstream performance without the need for additional data or pre-trained models.

## 1 Introduction

The importance of quality datasets in machine learning cannot be overstated. It is widely acknowledged that more data is better: model performance generally improves with data quantity, although such improvements often require exponentially more data (Kaplan et al., 2020; Udandarao et al., 2024) which might be running out (Sutskever, 2024). Fortunately, it has been shown that using higher data quality by data pruning leads to better scaling curves (Sorscher et al., 2022).

Current large models are trained on massive amounts of data collected from the Internet, which invariably contain many noisy examples. Methods grouped under the *data filtering* umbrella term involve data selection or pruning with the intention of removing noisy samples from datasets (Albalak et al., 2024). For large-scale datasets, these methods have typically employed heuristics such as filtering based on word count or image resolution, or using pre-trained models (Schuhmann et al., 2021; Marion et al., 2023) or additional data (Fang et al., 2023). Large models are increasingly often multimodal (Radford et al., 2021; Alayrac et al., 2022; Achiam et al., 2023; Gemini et al., 2024) and capable of linking information from distinct modalities like vision and language. Curating appropriate datasets is therefore a pressing issue, and in this work, we focus on filtering image-text pair datasets used to train CLIP (Radford et al., 2021) models.

A popular method for filtering vision-language datasets involves using a pre-trained model trained on the same task, for example, the OpenAI CLIP model, for selecting large image-text datasets (Schuhmann et al., 2021; Gadre et al., 2023). Although effective, this approach leaves unexplored some useful use cases and important scientific questions. First, it entangles the benefits of data selection with those of reusing and indirectly distilling the knowledge of the pre-trained model. Filtering methods should induce sample efficiency, but it's unclear if a filtering approach uses the samples more efficiently, given that we ignore the samples used to train the filter. Second, the selected dataset captures the capabilities but also the limitations and biases of the pre-trained models. For example, OpenAI's CLIP model is biased toward older events and exhibits lower retrieval performance of recent data (Garg et al., 2024). Third, having scientific inquiries on the nature of data filtering is harder to do when we don't have access to the pre-training dataset. These observations raise two questions. To start, can we do effective filtering *without relying on a pre-trained model*? Moreover, if we take into account the samples used to train the filtering model, does *filtering have any advantage in terms of sample efficiency*, i.e. does the filtering approach perform better given the same amount of samples seen? We will show evidence that both are possible.

Multimodal image–text pairs should describe the same semantic aspect; they should express the same story, idea, or scene, with the parts having high mutual information. Noisy samples are therefore those in which the visual and textual parts are mismatched. Meanwhile, many ML methods are learned to implicitly maximize a lower bound of the mutual information between input x and target y, e.g., by minimizing negative log-likelihood, or using contrastive losses (Hjelm et al., 2019). A good filtering method could estimate the alignment between an input and a target and select the maximal scoring samples. Of course, doing so in turn requires a model that can estimate the alignment between input and target and is already trained on valid samples, looping us back to the problem of noisy raw datasets. Thus, data selection and solving a task might be two facets of the same problem and might require the same level of understanding.

We'll devise a method for selecting data based on the memorization and generalization capabilities of ML models. Models learn *easy samples first*, and then memorize all data (Arpit et al., 2017). Large neural networks, in particular, can memorize noisy samples (Zhang et al., 2017). Usually, low-loss samples are considered easy for a model and are believed to be more likely clean. However, hard samples, which encode useful but non-trivial knowledge, require large capacity and long training times to be learned. Unfortunately, this is why, during training, these difficult samples typically incur high loss, which is also true for noisy ones. Herein lies a fundamental yet presently unsolved problem: *distinguishing hard samples from noisy samples*. At the same time, it intuitively motivates a simple strategy of making successive, increasingly informed decisions throughout training about what is noise and what is not. In this paper, we demonstrate the success of a Self-Filtering method in which a vision-language CLIP model is trained while also successively curating an evolving dataset. This is a form of self-improvement, where better models lead to better data mixes, which, in turn, improve the models.

**Contributions:**

- We show that effective filtering strategies can be achieved without using pre-trained models or additional data.

- We propose an image-text data selection method (termed *Self-Filtering*) consisting of iteratively training and improving the training dataset. Self-Filtering yields better datasets, which, in turn, produce better models.

- We suggest a strategy of mixing selected samples with random examples, balancing the exploitation of learned knowledge with the diversity available in the whole dataset.

- We show that, alongside a training run, filtering and downstream performance are correlated, leading to self-improving CLIP models.

## 2 Related work

Data quality is essential for training efficiency and improved performance by focusing on the most relevant samples, while ignoring noisy samples that can slow down and degrade the learning process. Since neural networks learn easy samples first, then memorize the rest of the data (Arpit et al., 2017), selecting samples with low loss is a good approach for preserving easy, likely clean samples (Schuhmann et al., 2021; Marion et al., 2023). On the other hand, other works suggest focusing on hard examples with high loss (Lin et al., 2017; Katharopoulos & Fleuret, 2018). Mindermann et al. (2022) balances learning on *easy* samples according to a reference model trained on a held-out dataset, and learning on *hard* examples according to the learning model. This approach is then scaled and made computationally efficient in (Evans et al., 2023) by inferring the complexity scores using smaller models.

**Self-Training and collapse:** Curriculum learning (Bengio et al., 2009) strategies suggest presenting data to a learning model in an ordered fashion, going from simple concepts to the more complex. Similarly, self-paced methods (Kumar et al., 2010) suggest constructing the order based on the current understanding of a learning model. In the proposed Self-Filtering method, we use a similar idea, of oversampling examples according to the current understanding of a model. But contrary to curriculum and self-paced methods, the induced order is not the key to the learning model. The model would benefit even more from a fixed but improved sample selection, but this might not be available during training.

The idea of self-training (Chapelle et al., 2009) has a long history in ML. It typically uses a pre-trained model to annotate an unlabeled dataset and fine-tune on confident samples (Lee et al., 2013; Xie et al., 2020). A downside is that the model will drift towards wrong but confident pseudolabels (Arazo et al., 2020). Similarly, a *model collapse* phenomenon is encountered in generative models (Shumailov et al., 2024; Herel & Mikolov, 2024) Nevertheless, it has been shown that the iterative training on self-generated data can be stable for visual generative models if the model has a good initialization and the generated data is mixed with real data (Bertrand et al., 2024; Arazo et al., 2020). In our context of filtering noisy vision-language samples, we select confident samples to oversample in training, while also following the ideas of mixing data from the real and filtered distributions and applying the filtering after a period of normal training.

**Self-Training for filtering:** An approach of alternating between filtering and retraining a model for small image datasets is proposed in Shen & Sanghavi (2019). They also show that the method recovers the ground truth in generalized linear models. Our work works in a similar vein, although we work in a large-scale setting of vision-language models, where we also incorporate the entire data. MentorNet (Jiang et al., 2018) is a filtering model used to assign importance weights to each sample during training. The downside is that it requires either using a fixed, pre-determined curriculum or it requires a hold-out dataset without noise to train the filtering network. Co-teaching (Han et al., 2018) trains jointly two models, where one model is trained on samples that the other model considers easy (i.e., have low loss). Although in time, the two networks will converge to the same model. Inspired by (Malach & Shalev-Shwartz, 2017), the Co-teaching+ method (Yu et al., 2019) applies co-teaching only on samples where the two models disagree. Self-training based on expectation-maximization is used to select generated samples for problem settings where the correctness of a sample can easily be evaluated (Singh et al., 2024).

**Vision-Language filtering:** Recent works focus on filtering large-scale visual-language datasets used for training models like CLIP (Radford et al., 2021). Xu et al. (2023) has shown that balanced data involving diverse concepts is essential and is one key ingredient for the original CLIP model. Datacomp (Gadre et al., 2023) benchmark assesses different methods for filtering noisy data for large image-text datasets. Their best filtering involves selecting image-caption pairs with high similarity according to a pre-trained CLIP model.

Alternatively, other works focus on selecting hard samples. Sorscher et al. (2022) clusters the data according to pre-trained embeddings and keeps a fixed number of hard examples (away from the centroid) for each cluster. Abbas et al. (2023) uses the same idea to remove near duplicates in the same cluster applied on LAION (Schuhmann et al., 2021) dataset. Then Abbas et al. (2024) keeps a varying number of samples from each cluster, depending on the cluster complexity. These works are based on the idea of removing

redundancies from the dataset and are complementary to removing noisy samples. They are also crucially dependent on pre-trained models to compute the embeddings for clustering.

Fang et al. (2023) use a set of high-quality, but proprietary datasets to train filtering models. They show that, in general, the performance of the filtering models for downstream tasks is uncorrelated with the performance of models trained on the datasets induced by the filtering models. Combining fixed datasets from distinct sources has been shown to produce models that have downstream performance in between models trained on individual sources (Nguyen et al., 2022). This is apparently contradictory to our method, where mixing likely clean and whole data helps, but in our case, one dataset is a subset of the other, thus the better properties of the likely clean set are also found in the larger one. Goyal et al. (2024) propose scaling laws that depend on the quality of the data in the context of CLIP models. Models using pre-trained models, can use the training CLIP loss (including negative pairs) for scoring the samples (Wang et al., 2024), or remove text from images to avoid relying on reading it (Maini et al., 2023) to achieve superior results.

**Similarity to desired task:** The current paradigm of pre-training on a large amount of internet-scale data, then using fine-tuning and few-shot or zero-shot prediction it is unclear which subsets of the data are relevant for the desired task. Xie et al. (2023) use hashed n-gram features to select samples similar to a subset of unlabeled downstream samples. Xia et al. (2024) selects relevant samples for instruction tuning by using a curated set of samples and computing similarities based on gradients. For vision-language filtering (Yu et al., 2023) also shows that selecting samples similar to the downstream evaluation is useful.

## 3 Self-Filtering

In the context of image-text contrastive pre-training, data filtering aligns with the training objective: that is, both aim to maximize alignment between the representations of the considered pairs (image-text). Training keeps the data fixed and optimizes the objective with respect to the model parameters, while data filtering keeps the model fixed and optimizes the same objective with respect to the data selection. This suggests a procedure for iterating between sample selection and model parameter optimization.

We consider the setting of large-scale datasets of image-text pairs $(x, y)$ collected from internet data (CommonCrawl, 2024). We train CLIP (Radford et al., 2021) models on this data, and test them on downstream tasks. The CLIP model $f_\theta$ is trained to minimize the distance between image and language embeddings corresponding to input pairs $h_x, h_y = f_\theta(x, y)$. When selecting samples for a new dataset, a common CLIP-based filtering that we use as a comparison is to select the pairs with the highest cosine similarity between their embeddings given by *pre-trained* OpenAI's CLIP ViT-L/14 model (Schuhmann et al., 2021).

What makes a filtering method based on a fixed model be effective? Is it the fact that the filtering model has more knowledge than the training method, for example by having additional data or longer training runs? Or does the filtering method better exploit the structure of the same data, with minimal additional information? To find this, we explicitly take into account the number of training samples seen (including repetitions) during training of the filtering model. A method is said to be more seen sample efficient than another if it reaches better performance using the same number of samples (including repetitions). Our goal is to produce an effective filtering approach that is more seen sample efficient when taking into account the samples used to train the filter. For this, we propose an iterative Self-Filtering approach. Aiming for this goal, we focus on two principal guiding notions:

**Observation 1: Noise vs hard samples.** The loss of a model is a good signal for distinguishing noisy samples from clean ones (Marion et al., 2023; Gadre et al., 2023). Nevertheless, using the loss of a single model at a given moment in time, it is notoriously challenging to distinguish hard samples from noisy samples.

**Observation 2: Exploitation vs diversity.** Using only the samples selected by a filtering model equates to exploiting its existing knowledge. Although a good idea for a few training steps, this is detrimental in the long term. For longer training times, we see the benefits of incorporating diverse samples from the entire distribution alongside likely clean samples.

---

**Algorithm 1:** Self-Filtering. Iterate between training the learning model $f_\theta$ for $T$ training steps and selecting a preferred, likely clean subset. Training is done on examples sampled from a mix of the complete dataset and the likely clean subset.

---

**1**      **Input**: Dataset $\mathcal{D}$ with $N$ examples, number of training steps T, number of rounds R, percent p
           **Result:** Trained model $f_\theta$; improved data mix $D_{current}$
**2**  // Phase 0: initialize model and data
**3**  initialize $f_\theta$
**4**  $\mathcal{D}_{current} \leftarrow \mathcal{D}$
**5**  **for** $round \in [1, R]$ **do**
**6**      // Phase 1: train on T x B examples uniformly sampled from the data mix
**7**      **for** $t \in [1, T]$ **do**
**8**         update $f_\theta$ using batch $\sim \mathcal{D}_{current}$
**9**      // Phase 2: select most probable samples
**10**     **for** $all\ (x, y) \in \mathcal{D}$ **do**
**11**        $h_x, h_y = f_\theta(x, y)$
**12**        $score(x, y) = $ cosine similarity $(h_x, h_y)$
**13**     create $\mathcal{D}_{likely-clean}$ of top p% scoring samples of $\mathcal{D}$
**14**     // Phase 3: new data mix: resample the whole dataset $\mathcal{D}$ with $\mathcal{D}_{likely-clean}$ having twice the weight
**15**     $\mathcal{D}_{current} \leftarrow$ resample N examples from $\mathcal{D} \uplus \mathcal{D}_{likely-clean}$

---

### 3.1 Proposed Self-Filtering Method

Guided by the previous observations, we propose a simple approach denoted Self-Filtering, shown in Algorithm 1. The method iteratively improves data filtering, balancing exploitation and diversity, all in a bootstrapped fashion without using pre-trained models or additional data. The main idea of the approach is to continuously train a learning model on increasingly better curated versions of the dataset. From time to time, the same learning model will act as a filter to select data that is more probable according to the information learned up to that point and then oversample this data. The method involves 4 main phases:

**Phase 0:** randomly initialize model $f_\theta$ and initialize the training data mix as the whole, unfiltered dataset.

**Phase 1:** train on examples sampled from the current data mix $D_{current}$ for $T$ training steps (train on $T \times$ batch-size examples). It is essential that the first time this phase is applied, the training phase is sufficiently long for the subsequent filtering step to make informative decisions

**Phase 2:** for all pairs $(x, y)$ in the dataset we compute a score signifying its credibility using the training model, acting as a filter. For vision-language models like CLIP, we simply use the cosine similarity between the embeddings of the visual and language parts. For classification, the negative of the training loss could be used (i.e., the log likelihood, or negative cross-entropy). We then select the top $p\%$ scoring samples.

**Phase 3:** we mix the selected subset back into the full dataset $\mathcal{D} \uplus \mathcal{D}_{likely-clean}$, where $\uplus$ denotes multiset union (concatenation). We then create a dataset of the same size as the initial one by resampling from the concatenation, effectively doubling the sampling weight of the likely samples.

We then repeat Phases 1-3, always using the current version of the data mix to continue training $f_\theta$.

Focusing only on the most likely samples poses the risk of converging to a biased set of easy samples that don't generalize. The same problem of accumulating errors and drifting towards a biased training set (Han et al., 2018) also appears in the related, and more general setting of self-training (Chapelle et al., 2009). This is why the third step of incorporating both likely clean and diverse data is essential to mediate this problem. It has been shown that generative models iteratively trained on their own outputs have two necessary conditions not to diverge: good initialization of the generating model, and the mixing of real data with the generated data (Bertrand et al., 2024). These conditions motivate the first and third phases of our iterative method. We do not assume that a model can reliably distinguish hard-but-useful samples

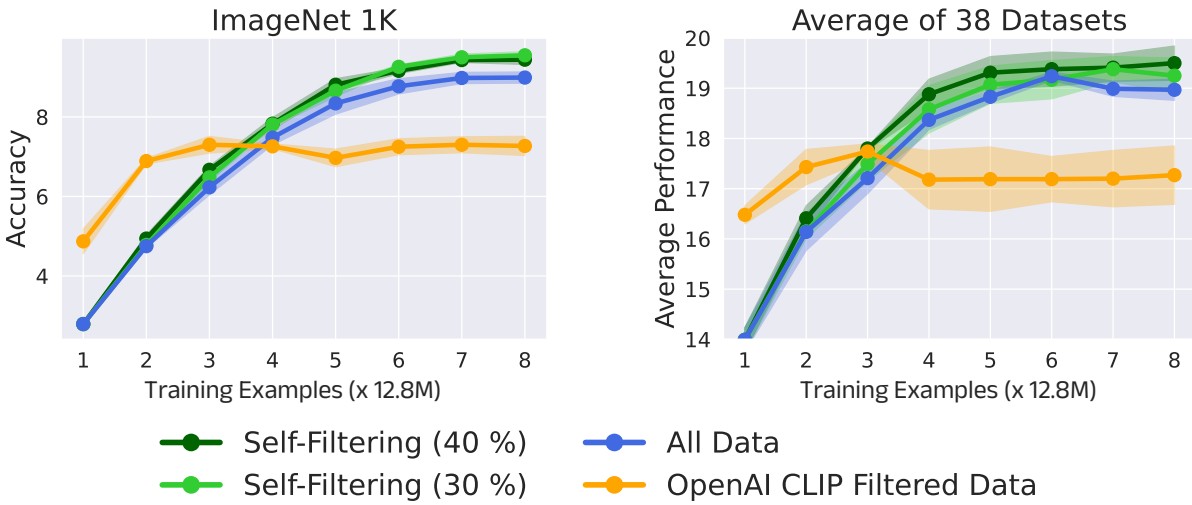

Figure 1: Datacomp small experiments trained to pass over $8 \times 12.8M$ examples (using 11.49M unique samples). We compare models trained by Self-Filtering (with mix of 30% or 40% selected samples) to models trained on either all data, or trained on data selected by OpenAI's CLIP. The proposed Self-Filtering approach leads to improved results. We show the average and $\pm 1$ standard deviation of 3 seeds.

from noisy samples early in training. Through the mixing phase, harder samples that are not selected in an early round are not permanently removed: they can still be observed through the unfiltered portion of the data mix, learned later, and selected in subsequent rounds, once the model has acquired the capability to understand them. This way, we avoid collapsing into easy samples by mixing the full distribution and reconsidering what is noise or not, achieving a good balance of exploitation and diversity.

## 4 Experiments

We test the filtering on the Datacomp (Gadre et al., 2023) testbed containing large amounts of text-image pairs. The benchmark is a standardized environment in which to compare different methods for selecting data to train OpenCLIP ViT-B/32 models (Ilharco et al., 2021), as well as evaluate the models' zero-shot performance on a set of 38 classification and retrieval tasks (e.g., ImageNet (Deng et al., 2009), ImageNetV2 (Recht et al., 2019), DTD (Cimpoi et al., 2014) and MSCOCO (Lin et al., 2014) etc.). In this work, we crawl the *small* scale subset of the dataset using the provided code and gather a set of 11.49M image-text samples from the 12.8M provided URLs, as the others are no longer available. We use the Datacomp code with default settings, if not otherwise specified, to train CLIP ViT-B/32 models and evaluate them zero-shot. We always train 3 seeds with different initializations and report the mean and 1 standard deviation of the metric for each task (e.g., accuracy or recall).

In the following, we begin by presenting the main results of our proposed Self-Filtering method in Section 4.1. We then present further results demonstrating the benefits of curating progressively enhanced datasets using successively better models in Section 4.2. Third, we show the benefits of a data mixture that emphasizes likely samples while still maintaining high diversity in Section 4.3. Additionally, we discuss the quality of the final mix produced by Self-Filtering Section 5.1, the relation to curriculum learning Section 5.2, and the correlation between a model's downstream and filtering performance Section 5.3. Finally, we provide the main limitations of the method in Section 5.4.

### 4.1 Main results of iterative dataset curation through Self-Filtering

We test the Self-Filtering approach and compare against baselines using the same number of gradient steps and using the same number of samples (including repetitions). The Self-Filtering model, shown in Algo-

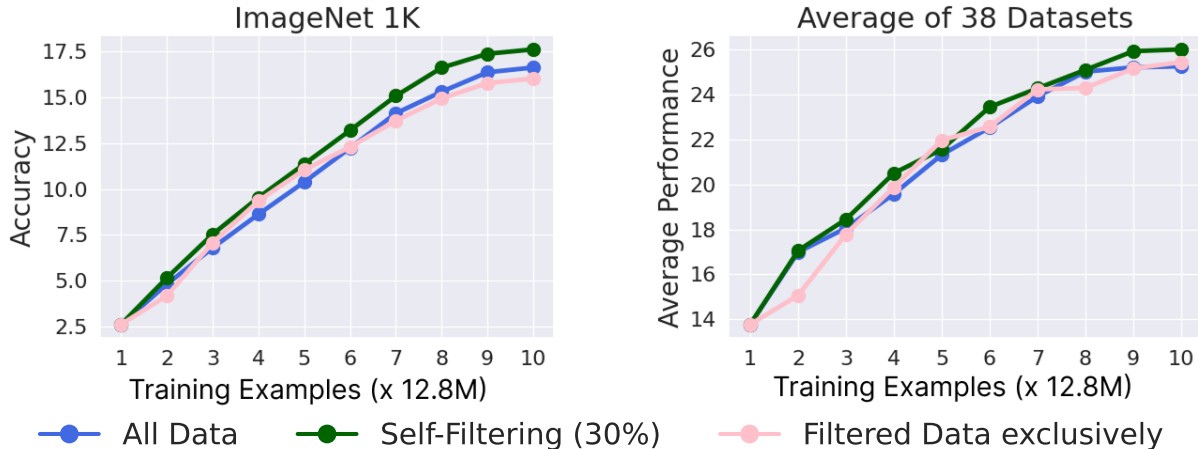

Figure 2: Experiments on the medium subset of Datacomp (128M unique samples). We apply the filtering operation every 12.8M samples to filter the next 12.8M subset of the data. We note that Self-Filtering improves over the baseline without filtering or training exclusively on self-selected data.

rithm 1, has $R$ rounds of training and data selection, where in each round the learning model $f_\theta$ is trained for $T$ steps.

**Results of Dataset Small subset.** For the Datacomp *small*, we choose $T$ such that at each round we train on a number of samples ($T \times$ batch-size $= 12.8M$) comparable to the initial dataset size ($D = 11.49M$). We use $R = 8$ rounds, resulting in approximately 8.9 epochs. Figure 1 compares our Self-Filtering approach against (1) a baseline of training on the complete dataset and (All Data) (2) training exclusively on the top 30% of samples filtered by OpenAI's CLIP ViT-L/14 (OpenAI CLIP Filtered Data). In all cases, we train all models using the same number of training steps. We train two Self-Filtering models, in which we create new data mixes in which either the top-30% or the top-40% of the most likely examples are oversampled. Both improve upon the baseline in downstream performance. See Figure 6 (Appendix) for other ratios. This shows that our simple Self-Filtering approach yields a higher-quality dataset, which, in turn, yields a better-performing model. This is especially important since Self-Filtering does not use any additional pre-trained model or additional data, yet achieves better performance.

In Table 1, we report the final checkpoints' performance of the previous models and two more baselines. Since training only on the OpenAI CLIP filtered data is detrimental in the long run, we also train on a mix of this subset and the entire data (OAI CLIP Filtered data + All) as we do for Self-Filtering. Finally, we get the final data mix obtained from Self-Filtering 30% and train a model from scratch on it (Final Self-Filtered 30% Data Mix). All models are trained for the same number of iterations. These last two models perform on par and show that the Self-Filtering approach can produce data as good as a strong pre-trained model.

**Results of Dataset Medium subset.** We perform Self-Filtering on the *medium* subset of Datacomp containing a total of $128M$ samples. Here, we use Self-Filtering in a streaming manner Algorithm 2, where we iterate over the dataset once and apply the training and selection of data in subsets (chunks) of the complete dataset. We apply the filtering procedure after seeing every $12.8M$ samples (R=10). At each filtering operation, we evaluate the next chunk of $12.8M$ examples and create a new mix of $12.8M$ examples where the top-30% scoring examples are sampled twice as much. In Figure 2 we see that using the Self-Filtering approach with the data mixing gives better performance than the baseline of using the entire dataset, as well as better than using only the $top - 30\%$ of the data, while all use the same number of training steps.

> **Takeaway.** Self-Filtering provides effective filtering leading to models with improved downstream performance. Data selected by Self-Filtering is on par with data selected by larger, pre-trained models.

Table 1: Results of CLIP ViT-B/32 trained on an available 11.49M out of 12.8M samples of Datacomp small for $8 \times 12.8\text{M}$ total examples seen. Self-Filtering method trained on an evolving data mix is competitive with a mix containing data selected by OpenAI's CLIP ViT-L/14 model, without requiring pre-training or additional data. If we train from the beginning on the final data mix obtained at the last round of Self-Filtering, we obtain performance on par with filtering by OpenAI's CLIP ViT-L/14, showing the quality of the data produced by Self-Filtering.

| Data | ImageNet | ImageNet Shifts | VTAB | Retrieval | Average |
|---|---|---|---|---|---|
| All data | $9.0_{\pm 0.1}$ | $8.2_{\pm 0.2}$ | $18.9_{\pm 1.1}$ | $14.1_{\pm 0.6}$ | $19.0_{\pm 1.1}$ |
| OAI CLIP Filtered data | $7.3_{\pm 0.2}$ | $7.2_{\pm 0.3}$ | $18.3_{\pm 1.0}$ | $10.2_{\pm 0.2}$ | $17.3_{\pm 0.9}$ |
| OAI CLIP Filtered data + All | $9.8_{\pm 0.0}$ | $9.1_{\pm 0.2}$ | $19.6_{\pm 0.9}$ | $14.4_{\pm 0.7}$ | $19.6_{\pm 1.1}$ |
| Self-Filtering 40% | $9.4_{\pm 0.1}$ | $8.7_{\pm 0.3}$ | $\mathbf{19.7}_{\pm 1.3}$ | $14.2_{\pm 0.3}$ | $19.5_{\pm 1.0}$ |
| Self-Filtering 30% | $9.6_{\pm 0.1}$ | $8.7_{\pm 0.3}$ | $18.9_{\pm 1.3}$ | $14.5_{\pm 0.4}$ | $19.2_{\pm 1.1}$ |
| Final Self-Filtered 30% Data Mix | $\mathbf{9.9}_{\pm 0.1}$ | $\mathbf{9.3}_{\pm 0.3}$ | $19.1_{\pm 1.4}$ | $\mathbf{14.6}_{\pm 0.4}$ | $\mathbf{19.7}_{\pm 0.9}$ |

## 4.2 Investigation: Iteratively selected datasets yield progressively better models

Previously, we trained a model continuously on the evolving dataset. The performance gains could be attributed to the increasing quality of the data, but could also be due to other regularisation effects independent of data quality. We design an experiment to assess whether dataset quality does indeed improve across iterations. Specifically, we explore a scenario in which we create a few increasingly better curated datasets and use each one to train a separate model from scratch. This allows us to compare the quality of each dataset according to the performance of the model trained on it.

We train a CLIP model (CLIP-Round-1) from scratch on the small-scale Datacomp dataset for a fixed number of steps corresponding to seeing 12.8M samples as recommended by Datacomp. We then use this model to filter the entire dataset and train a new model from scratch using this new subset for the same number of steps, corresponding to 12.8M samples, yielding the second model, CLIP-Round-2. Table 2 shows that CLIP-Round-2 improves over the CLIP-Round-1. We repeat the same procedure multiple times, each time using the previously trained model to filter a new dataset from the whole data and use it to train the next model from scratch. Iterating on training and filtering sequentially improves the performance of the resulting models. These results support the *hypothesis that better models filter better datasets, and that these datasets can in turn train better models.*

We emphasize that training multiple rounds of models solely to obtain a better filter is computationally expensive, and this budget could instead be used to simply train a model on the entire dataset for longer. We observed that if we train a single model on the whole dataset for $T \times R$ steps, instead of starting from scratch $R$ times, it would produce a better model (CLIP-longer from Table 2). This suggests that resetting the model and starting from scratch each round is wasteful. Thus, it motivates our proposed Self-Filtering, in which a single model is continually trained on successive rounds of improved datasets and produces better results with the same number of training steps.

> **Takeaway.** Models trained from scratch on iteratively better curated datasets improve on each other. These results suggest that enhanced models, which in turn filter improved datasets, contribute to the success of our method.

## 4.3 Investigation: Balancing exploitation and diversity by mixing data subsets

The sequence of decisions for sample selection profoundly influences the path that a learning model takes through parameter space. Selecting data samples preferred by a pre-trained model effectively equates to *exploiting* the existing knowledge of that model. Discarding data potentially excludes data containing in-

Table 2: Average performance on the 38 tasks of the Datacomp benchmark at the *small* scale. We train from scratch multiple models, each one trained on data filtered by the previous model. Each successive model is better than the previous, suggesting that the produced data mix is improving. Models trained on this data obtain similar performance to models trained on data selected by a strong pre-trained model such as OpenAI's CLIP ViT-L/14. Nevertheless, using all the training compute for a single training run (CLIP-longer) is better.

| Training Model | Unique samples | Samples Seen | Filtering Model | Avg Performance |
|---|---|---|---|---|
| CLIP-longer | 11.49M | $4 \times 12.8$M | none | $18.4 \pm 0.9$ |
| CLIP-Round-1 | 11.49M | 12.8M | none | $13.5 \pm 0.3$ |
| CLIP-Round-2 | $0.3 \times 11.49$M | 12.8M | CLIP-Round-1 | $14.6 \pm 0.4$ |
| CLIP-Round-3 | $0.3 \times 11.49$M | 12.8M | CLIP-Round-2 | $15.4 \pm 0.3$ |
| CLIP-Round-4 | $0.3 \times 11.49$M | 12.8M | CLIP-Round-3 | $\mathbf{15.6} \pm 0.3$ |
| CLIP-oai-filtered | $0.3 \times 11.49$M | 12.8M | OpenAI-CLIP | $\mathbf{15.8} \pm 0.2$ |

formation not yet learned by the model. While it's true that a goal of filtering is to omit noisy data, a tradeoff must be achieved between including hard data points and noisy data points if the next model is to learn new information. The problem is evident when we consider filtering a dataset that includes events that are more recent than the pre-trained model. It has been shown that OpenAI's CLIP struggles with retrieval tasks involving recent concepts such as Covid-19, or the game Wordle Menon & Vondrick (2022); Garg et al. (2024) that happened after the model was trained. This motivates *exploration* of data subsets that are unlikely or unfamiliar to the older model.

We trial a simple balancing between the data preferred by a pre-trained model and the whole dataset. We train CLIP ViT-B/32 from scratch for $8 \times 12.8$M steps. Unlike before, where we used a cosine learning rate scheduler, here we use a constant learning rate to disentangle the relationship between the learning rate and the moment when we filter the data. We then mix the whole dataset with the top-30% of samples preferred by OpenAI's CLIP ViT-L/14 model and train a new model (Fix Data Mix: OpenAI CLIP Filtered + All) on this dataset. We compare against training from scratch models on all data (All Data) or only on the selected subset (OpenAI CLIP Filtered Data). Figure 3 a) shows that when training a model from scratch, the mixing strategy achieves the best results, showing that this simple strategy is effective at making use of the information in both the subset and the wider, diverse dataset. Figure 3 b) shows models initialized from the checkpoint OpenAI CLIP Filtered Data (from the first panel) from the training step corresponding to $1 \times 12.8M$ examples seen, and then continued trained on either the data mix or the whole dataset. Figure 3 c) shows models initialized from the step corresponding to $4 \times 12.8M$ samples seen of CLIP Fix Data Mix model (from the first panel) and continued training on either all data or selected data. This shows that training on the data mix from the start gives a good overall model.

> **Takeaway.** Training on a mix of selected and all data improves over either using only preferred data or using all data, achieving a good balance of exploring novel, diverse samples and exploiting samples preferred by the filtering model.

# 5 Discussion

## 5.1 Self-Filtering produces datasets as good as large pre-trained models.

We also highlight that both the fixed datasets selected by the Self-Filtering approach and OpenAI's CLIP model produce models that achieve similar downstream performance (see Figure 4). This proves that Self-Filtering produces datasets comparable in quality to pre-trained models.

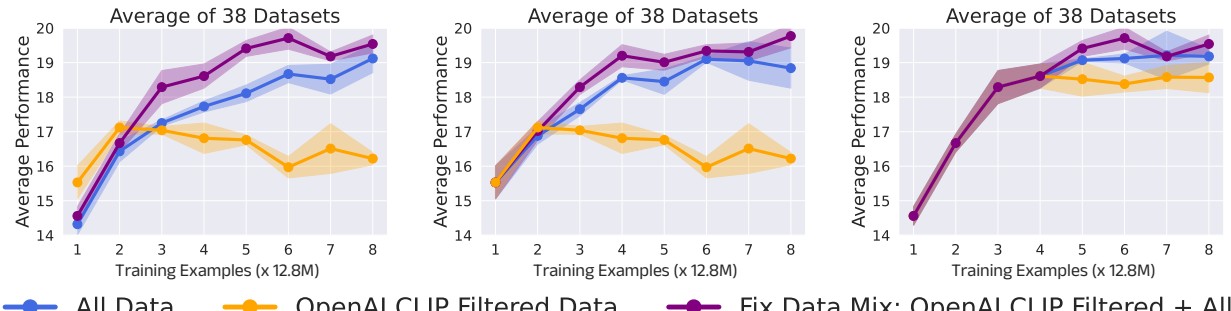

Figure 3: Comparison of CLIP models trained on different data subsets: the entire data, exclusively the data filtered by OpenAI's CLIP, or a mix of all and the same filtered data. In the second and third panel the models are initialized from the same checkpoint from steps corresponding to 12.8 M and $4 \times 12.8$M examples seen, respectively. For a small number of training steps, using exclusively filtered data is best, but overall, the mixing strategy achieves the best results. Unlike before, we use a constant learning rate here.

**Transferring Self-Filtering.**   Our main experiments on Datacomp small apply Self-Filtering over multiple epochs. We explore whether the Self-Filtering models can filter noisy datasets unseen during training. This transfer property of a filtering model would be essential in the widely used one-epoch training regime. To this end, we take the final model obtained by Self-Filtering on the small subset of Datacomp and use it to filter another *distinct* subset of Datacomp medium that has a similar size (12.8M samples in total). In Figure 5, we see that training on a mix of the complete dataset and the selected dataset achieves better results than the baseline trained on the whole dataset and has performance similar to training on a mix containing samples filtered by OpenAI's CLIP. This proves that the models obtained by Self-Filtering transfer to new datasets and can effectively filter samples unseen during training.

## 5.2   Benefits of Self-Filtering are not due to curriculum learning.

When doing Self-Filtering, we implicitly define a curriculum of the data. This is due to selecting likely clean samples as defined by an evolving model. Since what is considered easy is evolving with the model, we can view the data as evolving from simple to slightly more complex, thus having a curriculum. Nevertheless, we will see that there is no benefit for the training model from receiving data in a curriculum fashion. Instead, the learning model would benefit from the most accurate data, but our estimate of the most accurate data evolves as we improve the model, thus creating a non-stationary dataset. To validate this, we train a new model from scratch (Fix Data Mix: Self-Filtered Data + All) using the data from the last Self-Filtering iteration of a previously trained model. As seen in Figure 4, this model trained on a fixed dataset is better than the original Self-Filtered model. This proves that the key is to have the right data, and not the curriculum implicitly defined by the data selection.

## 5.3   Downstream and filtering performance. Correlated or independent?

Self-Filtering is based on the bootstrapping hypothesis that a model can produce a better dataset that, in turn, can produce better models. This suggests that filtering performance is correlated with downstream performance. This apparently contradicts previous works like (Fang et al., 2023) that state that the two are not correlated. Nevertheless, in our approach, we don't claim that *any* models should have these performances correlated. We only show that checkpoints from the same training run have filtering and downstream performance correlated. Moreover, we show that the data obtained by Self-Filtering using a smaller and less performant CLIP ViT-B/32 is as good as data filtered by the bigger and more performant OpenAI's CLIP ViT-L/14 Table 1, Figure 4, Figure 5. In general, we need further investigations into the correlation of filtering and downstream performance of different models trained independently on different data, but for the same model at different time steps, the two are correlated, as our results show.

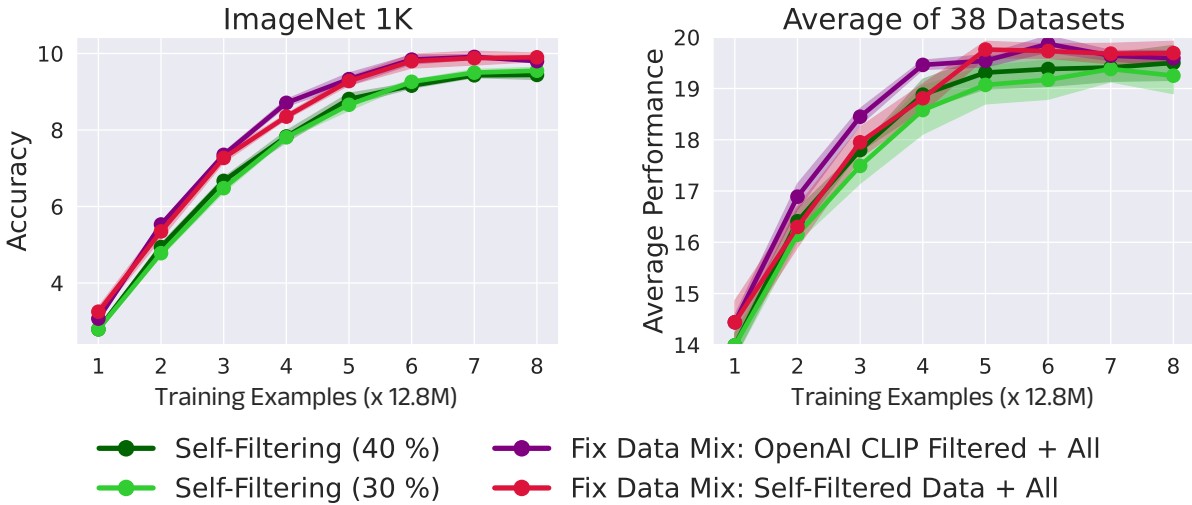

Figure 4: Runs on Datacomp small. We compared a model trained on a mix of all data and data selected by a previous Self-Filtering (30%) method. This data mix achieves similar performance to the data mix created by OpenAI's CLIP ViT-L/14 model, proving that Self-Filtering selects good datasets, even without pre-training or additional training data.

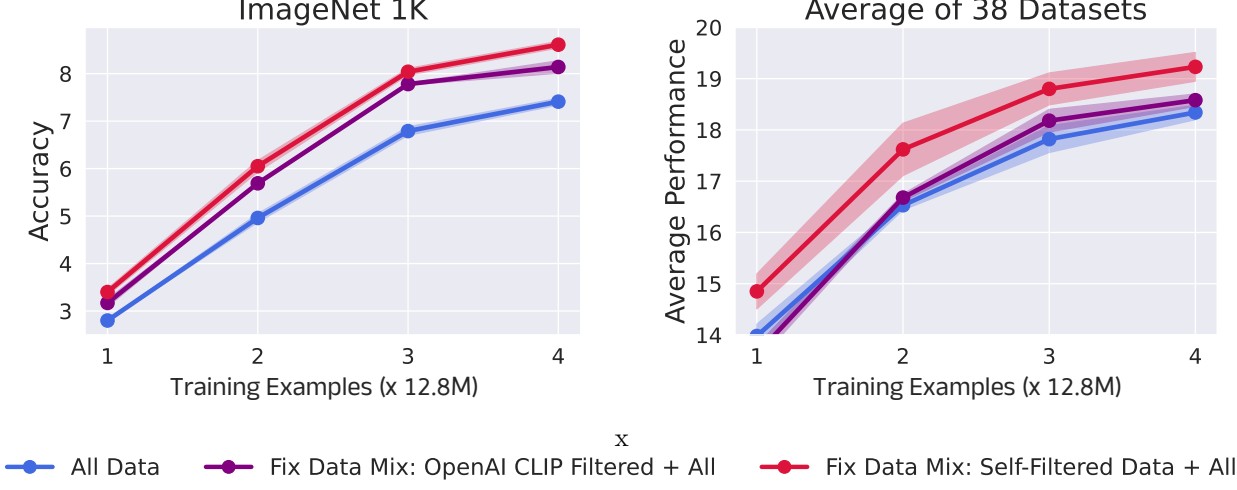

Figure 5: Transfer experiments on a subset of Datacomp medium. A model trained with Self-Filtering on the small Datacomp is used to filter a Datacomp medium subset. The model trained on the dataset obtained with Self-Filtering achieves the best results, surpassing the model trained on all data or data selected by OpenAI's CLIP. This proves that models trained by Self-Filtering can be used to filter data unseen during training.

Additionally, Evans et al. (2024) trains 5 different models using a large number of examples seen (from 250M to 3B), and they observe "a striking correlation" between filtering and downstream performance in their setting, where the filtering models are trained on data from the same distribution. The observations of both works fit well with our work and give a clear and important motivation for our method on Self-Filtering.

### 5.4 Limitations

An ideal filtering method will assess the usefulness of the selected data for downstream tasks. Our method uses the cosine similarity for estimating the importance, but this might not be aligned with the final goal. We

only assume that patterns that are consistently learned across the training run are relevant for downstream tasks. Moreover, the decision is still local, without considering second-order interactions between samples and without looking at the future model or downstream task.

Our analysis is on Datacomp small (12.8M samples) and medium (128M samples), which could benefit from validations at a larger scale, in terms of the number of unique samples and training steps.

One important note is that the analysis, as it is done in the Datacomp setup (Gadre et al., 2023), does not take into account the time spent scoring the samples for selection. This computational budget could instead be used for training on the entire dataset, without data selection. In practice, because inference is faster and uses less memory, scoring the data could be done in parallel on older and smaller devices that are not actively used for training purposes.

Self-Filtering mitigates the problem of distinguishing noise from hard samples by selecting more difficult samples later in training as the model evolves. Second, through the mixing strategy, we ensure that we always have access to hard examples, balancing exploitation and diversity. Both problems are yet to be fully solved.

## 6 Conclusion

Data filtering is a challenging task, having at its core the problem of distinguishing hard examples from noisy ones. We have proposed a simple approach towards tackling it, using a Self-Filtering method that uses the same model to learn, as well as selecting likely clean samples. Our method iterates between training a model on the current dataset and using the model to select likely clean samples to curate its next training dataset. We validate that a straightforward blend of likely clean and unfiltered data can provide a good balance between exploiting the learned knowledge at that point and making use of the data diversity. We empirically demonstrate the capabilities of the method to effectively filter image-text datasets, all without using any pre-trained models or additional data.

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

## A    Broader Impact Statement

This paper involves filtering vision-language data and training associated CLIP models. CLIP models are widely used as components in larger ML systems, such as retrieval, captioning, image generation, and large vision-language models. Any biases learned by the backbone CLIP models can affect the final downstream system. Potential concerns include sensitive information included in the training dataset, such as identifying or personal information, and sensitive topics. The data might contain harmful or stereotypical representations related to protected characteristics like race, gender, and sexuality. Since the paper involves data filtering, biases in the data are crucial. Pretraining models carry over biases from their respective training data; thus, by avoiding them, we can better control the behavior of new models. Conversely, this increases the importance of the current training data and how we filter it.

A second layer that can potentially introduce or exacerbate biases is the training algorithm. Most ML algorithms are based on average performance and are biased to the majority views. This could introduce bias towards popular or hegemonic views, leading to degraded performance for minority views or harmful consequences, depending on the application.

Using a bootstrapped approach like Self-Filtering tends to select samples that confirm the current understanding of the model. This could lead to a collapse into a self-confirming set of samples and views. As discussed in the paper, we address this self-confirming collapse through the data mixing strategy. While the complete collapse is avoided, in practice, we could only expect amelioration, not a complete fix.

Further mitigation strategies could include better balancing of the data from the outset, in terms of minority concepts, cultural practices, language variety, or geographic locations. Additionally, applying different levels of filtering for different subgroups could also help in maintaining diversity. For example, different thresholds might be used for different subgroups. While this is desirable, we also need to have access to subgroup partitioning, which is non-trivial. We encourage further research in this direction.

## B    Appendix

### B.1    Streaming Algorithm

In cases where the training dataset is big, we apply the Self-Filtering algorithm in a streaming fashion in chunks of the dataset of size $T \times$ batch-size Algorithm 2.

### B.2    Transfer results on a subset of Datacomp medium

As explained in Section 5.1, we create a subset of Datacomp medium containing 12.8M samples that is completely disjoint from Datacomp small. We then filter this Datacomp medium subset with a pre-trained CLIP model or with a model trained on Datacomp small via Self-Filtering. In Table 3 we present the complete results showing that the filtering with the model produced by Self-Filtering is best, improving over OpenAI CLIP filtering. This suggests that models trained by Self-Filtering transfer can be used to effectively filter unseen samples.

### B.3    Ablation: Top-p Self-Filtering

We apply our Self-Filtering method with varying percentage $p$ for selection and show the results in Figure 6. We can see that selecting the top-30%/40% of samples is best.

### B.4    Results for each downstream task

Figure 7 and Figure 8 show results on each individual downstream tasks.

---

**Algorithm 2:** Streaming Self-Filtering. Here, a big dataset $\mathcal{D}$ is processed in non-overlapping chunks. We iterate between training the learning model $f_\theta$ for $T$ training steps on the current chunk and selecting a preferred, likely clean subset of the next chunk. Training is done on a mix of the likely subset and the whole next dataset chunk.

---

1       **Input**: Dataset $\mathcal{D}$, number of rounds R, number of steps T, percent p **Result:** Trained model $f_\theta$; improved data mix: union of all $D_{current}$ sets

2  // Phase 0: initialize model and get the next chunk of data

3  initialize $f_\theta$

4  $\mathcal{D}_{current} \leftarrow$ next chunk from $\mathcal{D}$ of size $T \times B$

5  **for** $round \in [1, R]$ **do**

6      // Phase 1: train on T x B examples uniformly sampled from the data mix

7      **for** $t \in [1, T]$ **do**

8         update $f_\theta$ using batch $\sim \mathcal{D}_{current}$

9      // Phase 2: select most probable samples of the next data chunk

10      $\mathcal{D}_{next} \leftarrow$ next chunk from $\mathcal{D}$ of size $T \times B$

11      **for** $all$ $(x, y) \in \mathcal{D}_{next}$ **do**

12         $h_x, h_y = f_\theta(x, y)$

13         $score(x, y) = \text{cosine sim } (h_x, h_y)$

14      create $\mathcal{D}_{likely-clean}$ of top p% scoring samples of $\mathcal{D}_{next}$

15      // Phase 3: new data mix: resample the next chunk $\mathcal{D}_{next}$ with $\mathcal{D}_{likely-clean}$ having twice the weight

16      $\mathcal{D}_{current} \leftarrow$ resample $T \times B$ example from $\mathcal{D}_{next} \uplus \mathcal{D}_{likely-clean}$

| Data | ImageNet | ImageNet Shifts | VTAB | Retrieval | Average |
|------|----------|-----------------|------|-----------|---------|
| All data | $7.4_{\pm 0.1}$ | $7.1_{\pm 0.2}$ | $18.0_{\pm 1.0}$ | $14.7_{\pm 0.3}$ | $18.3_{\pm 0.8}$ |
| OAI CLIP Filtered data + All | $8.1_{\pm 0.1}$ | $7.8_{\pm 0.2}$ | $18.7_{\pm 1.3}$ | $14.9_{\pm 0.2}$ | $18.6_{\pm 0.9}$ |
| Self-Filtered CLIP + All | $\mathbf{8.6}_{\pm 0.1}$ | $\mathbf{8.2}_{\pm 0.2}$ | $\mathbf{19.0}_{\pm 1.8}$ | $\mathbf{14.8}_{\pm 0.2}$ | $\mathbf{19.2}_{\pm 1.0}$ |

Table 3: Results of CLIP ViT-B/32 trained on a subset 12.8M samples of Datacomp medium for $4 \times 12.8M$ steps. Self-Filtering method is competitive with selection by OpenAI's CLIP ViT-L/14 model, without requiring pre-training or additional data. A model trained on the final dataset obtained by Self-Filtering, obtains performance on par with filtering by OpenAI's CLIP ViT-L/14.

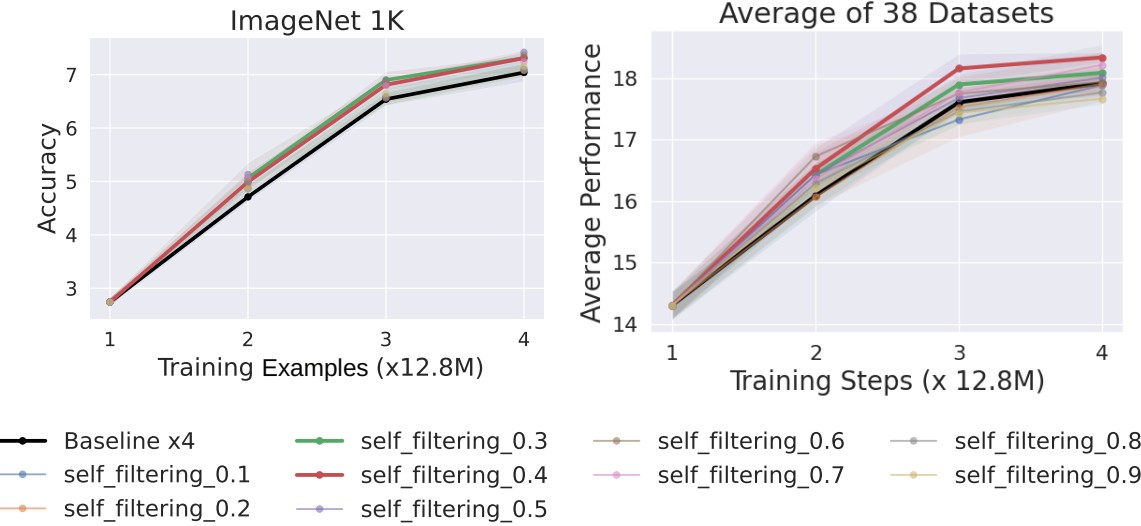

Figure 6: We ablate the percentage of top-$p$% samples selected in the Self-Filtering method. For computational reasons, we only train the models for a number of steps corresponding to seeing $4 \times 12.8M$ examples. We observe that keeping $30-40\%$ of the samples gives good performance compared to the baseline of training on the complete dataset

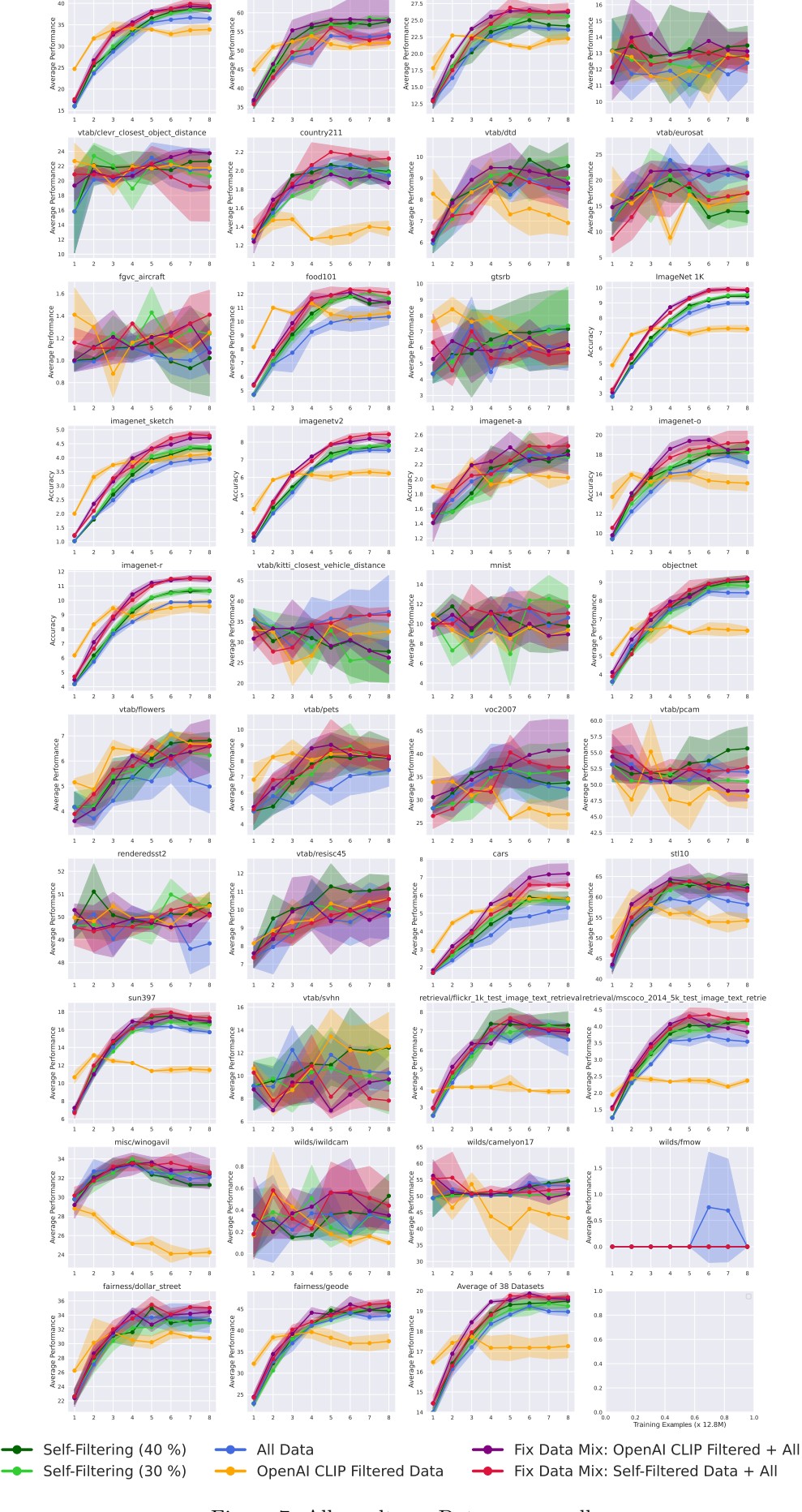

Figure 7: All results on Datacomp small

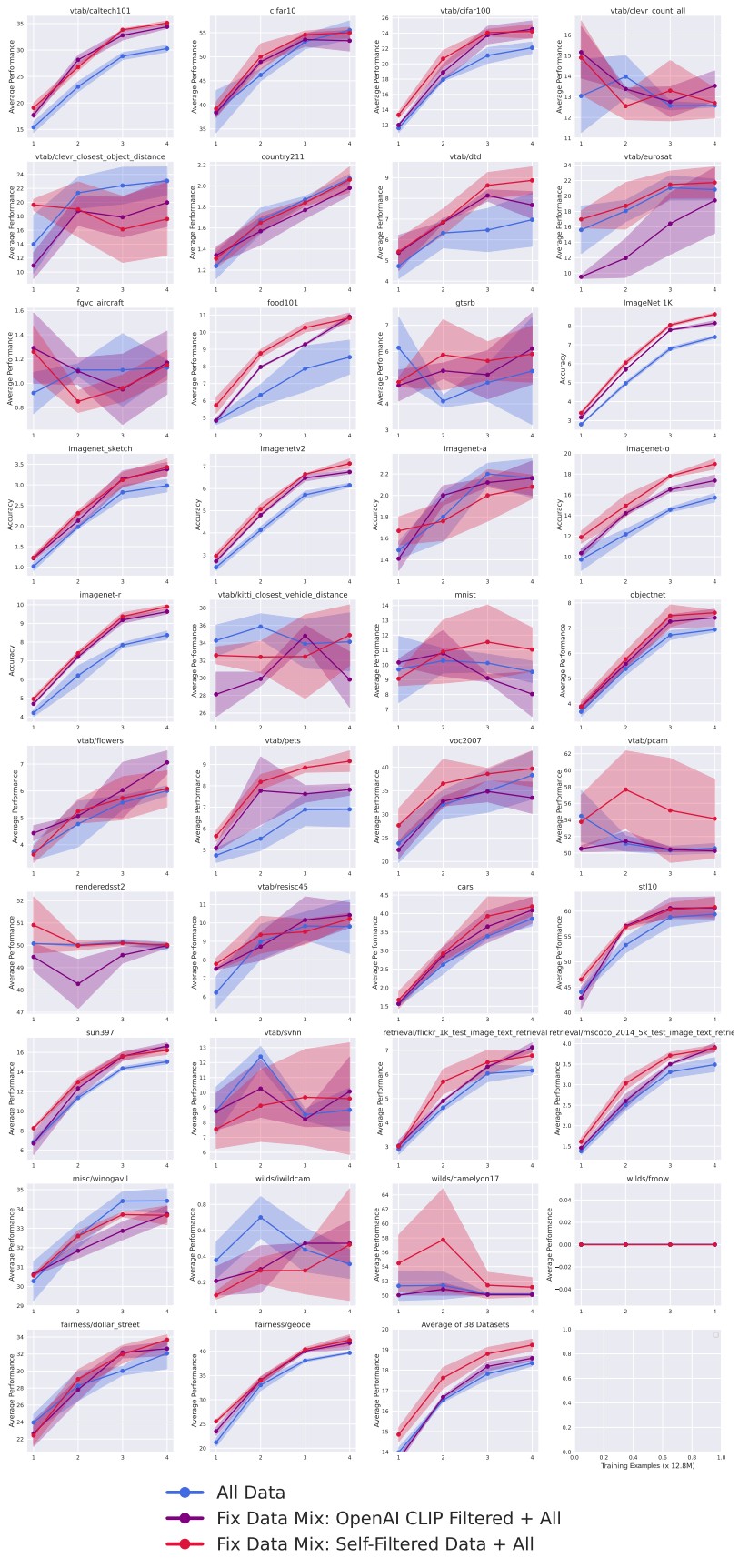

Figure 8: All results on Datacomp medium subset of $12.8M$ samples

