# OpenReview forum: "Data Selection Through Iterative Self-Filtering for Vision-Language Settings"
_TMLR — Accepted by TMLR_

### Review · Reviewer_GfV7 · 2026-03-08

**Summary Of Contributions:**

Summary

High-quality data is one of the most important factors in training neural networks. In practice, the noisy data can deteriorate the performance of the models but existing methods for filtering these noisy data often rely on a pre-trained model which may not be available. This paper proposes Self-Filtering, a data selection method for vision-language data. It works by iterating between model training and data selection to construct a self-evolving high-quality dataset. This is based on the observation that both the training and data filtering have a common objective of maximizing the alignment between the image and the text. Empirical evaluations show the effectiveness of the proposed method.

Strengths
- The proposed method shows that obtaining a high-quality vision-language dataset is possible without a good pre-trained model.
- The paper runs extensive ablation experiments from Sections 5.1 to 5.3, which provide insights into how the proposed method works.
- The paper acknowledges its limitations from various perspectives in Section 5.4.

Weakness
- The quality of the presentation is poor and each paragraph does not effectively convey its main idea. The paper also contains several grammatical errors (e.g., singularity vs plurality), latex errors, and lacks coherence.
- The paper argues that existing methods relying on a pre-trained model have disadvantages because pre-training the model can be costly and biases of the pre-trained models can be transferred. However, the paper does not clearly demonstrate that these disadvantages are effectively addressed with the proposed method.

**Additional Comments:**

None.

**Audience:**

Yes

**Audience Explanation:**

The importance of constructing the high quality data is becoming more and more significant as the training costs become larger. The paper considers a timely topic in which lots of the audience working in the related area would be interested.

**Broader Impact Concerns:**

The paper does not seem to have broader impact concerns.

**Claims And Evidence:**

No

**Claims Explanation:**

In Section 1 and the third paragraph of Section 3, the paper criticizes the filtering methods which involve using a pre-trained model in that (1) the biases of the pre-trained models can be inherited and (2) pre-training the model can be costly. However, it is unclear if the proposed method effectively addresses these issues. For (1), the paper does not show that (i) the methods relying on pre-trained models indeed possess biases present in the pre-trained models and (ii) these biases are not present for the models trained with Self-Filtering. For (2), the paper does not compare its method with baselines (All Data and OpenAI Filtered Data) in terms of costs (e.g., FLOPs or runtime). Specifically, the performance gain of Self-Filtering compared to All Data seems quite marginal and thus the proposed method might be ineffective when considering the cost for curating the data.

**Requested Changes:**

(major) The paper should compare the proposed method with baselines in terms of runtime or FLOPs to argue that it remains efficient compared to others. Here, the runtime should include all the costs for training the models (e.g., training cost for All Data vs training cost + filtering cost for Self-Filtering).

(major) If the paper criticizes the previous methods relying on pre-trained models for their potential risk of exposure to biases, the related experiments should be included to demonstrate that (i) the previous methods indeed inherit the biases present in the pre-trained models and (ii) Self-Filtering addresses this issue.

(major) The presentation of the paper should be significantly improved. Overall, the paper lacks clarity, precision, and coherence. Also, there are lots of incorrect LaTeX expressions and grammatical errors. For example, the paper incorrectly uses \citet where it should change to \citep (i.e., with parentheses) and the last sentence of the caption of Figure 1 is sliced.

---

> ### Author Response · Authors · 2026-04-04
> **Rebuttal**
>
> We thank the reviewer for their insightful comments and suggestions, and we respond below.
>
> ## Computational efficiency
>
> Please see the “Sample efficiency vs. computational efficiency”  section in the general response above.
>
>
> ## Biases of the pre-trained model
>
> The main goal of this paper is to investigate the filtering capabilities in the absence of a pre-trained model. It is unclear whether we can use a model to filter its own data in the context of vision-language models. We research this aspect and consider it a worthy scientific investigation. Beyond the scientific endeavor, there are practical aspects that we cannot use or want to avoid a pre-trained model.
>
> One reason not to rely on pre-training models is that they can reproduce the biases present in their training data. These biases can take many forms: the data might be mostly old, contain few recent samples, or originate from a specific geographic location, language, or culture. In the paper, we mention Garg et al. (2024), who provide evidence that OpenAI’s CLIP model is biased toward older events and has lower retrieval performance for recent data. Analyzing the biases of datacomp and models trained on it compared to the original OpenAI model is beyond the scope of this paper. We simply point out that if a pre-trained model is trained on more biased data, it will inherit those biases. Sometimes, preventing this is desirable, which is why training a new model on a fresh dataset is recommended.
>
> ## Presentation of the paper
>
> As suggested by the reviewer, we rephrased some sentences, corrected several typos, and fixed the citations and the slice shown in Figure 1. We would be glad to revise any specific paragraphs or sections that remain unclear or could benefit from additional explanations.

---

> > ### Comment · Reviewer_GfV7 · 2026-04-06
> > **Reply to rebuttal**
> >
> > I thank the authors for the detailed response. I agree with the authors that "*there are practical aspects that we cannot use or want to avoid a pre-trained model.*". My major concern was that this scenario was mainly assumed in the first version for (1) when the pre-trained model contains biases and (2) it requires huge cost for pre-training the models, which was not explicitly addressed with the proposed method. While the authors' rebuttal has mostly addressed this concern, I encourage the authors to make it clearer in the revised version about the computational efficiency, biases of the pre-trained model, and other scenarios where we cannot use or want to avoid a pre-trained model.

---

### Review · Reviewer_N7Be · 2026-03-16

**Summary Of Contributions:**

The paper proposes an iterative self-filtering method for curating image-text datasets used to train CLIP models. The core idea is to train a CLIP model, use its cosine similarity scores to identify the top-p% likely clean samples, mix them back into the full dataset (doubling their sampling weight), and repeat. Hence, no external pretrained model or additional data is needed.
The proposed method is evaluated on Datacomp small (12.8M) and medium (128M) subsets. Results show Self-Filtering improves over training on all data and matches the performance of filtering with OpenAI's CLIP ViT-L/14, a much larger pretrained model.
In addition, the authors propose a streaming variant (Algorithm 2) for larger datasets that cannot be revisited in their entirety.
The authors also perform detailed ablations that measure whether iterative rounds produce progressively better datasets (Table 2), whether the mixing strategy outperforms filtered-only or all-data training (Figure 3), and rule out curriculum learning as to why they perform well (Section 5.2), and the sensitivity of the proposed approach to the selection percentage p.

Strengths:
- The problem setting is practical and well-motivated. In practice, data filtering methods often rely on additional models to select training instances, which has several issues: (i) the biases of the pretrained filter model are inherited by the new model, and an appropriate model might not exist for your specific task or domain.
- The method is simple and easy to implement.
- The experimental design is careful in some places, particularly the experiment that trains the model from scratch on the final data mixture (Section 5.2), which disentangles data quality from curriculum effects.
- Running 3 seeds with std reported throughout ensures robustness of inferences.

Weaknesses:
- The algorithmic novelty over Iterative Trimmed Loss Minimisation (Shen & Sanghavi, 2019) is narrow. The mixing step is the key difference.
- Experiments are limited to small and medium Datacomp scales. The paper does not compare against several strong recent baselines on the Datacomp benchmark.
- The computational cost of scoring the entire dataset at each round is acknowledged but not quantified, and hence, the efficiency claims are hard to evaluate.
- Confirmation bias from single-model self-selection is a known failure mode (the Co-teaching literature exists precisely because of this), and the paper does not discuss it.

**Audience:**

Yes

**Audience Explanation:**

- Data curation for VLMs is an active and important research area, and any method that avoids dependence on pretrained filter models addresses a real, practical need.
- The bootstrapping idea (model improves data, data improves model) is conceptually appealing and connects to broader themes around self-improvement and model collapse that the community cares about.

**Broader Impact Concerns:**

The paper does not include a broader impact statement, and it should be added. Self-Filtering inherits whatever biases the model develops during training and amplifies them through iterative selection. If the initial training phase leads the model to underrepresent certain demographic groups, visual styles, or geographic regions, the filtering loop will systematically downweight data from those categories in subsequent rounds. This is a concrete risk that should be acknowledged.

**Claims And Evidence:**

Yes

**Claims Explanation:**

The claims made in the paper are only partially supported, here are the specific issues:
- The central claim that Self-Filtering matches OpenAI CLIP filtering is well supported at Datacomp small  (Table 1, Figure 4). The numbers are close, the comparison is fair (same training budget), and hence the comparison is convincing.
- The medium-scale experiment (Figure 2) uses a streaming variant that is structurally different from Algorithm 1. The comparison here is only between all data and filtered data, not between OpenAI CLIP filtered + all, and this weakens the medium-scale claim.
- The paper claims that self-filtering works without investing significant budget into training a filtering model, but the method requires scoring all N samples at every round. With R=8 rounds, that is 8 full forward passes over the dataset in addition to training. The paper mentions this as a limitation but does not provide wall-clock times or FLOP counts. Without this, the efficiency claim is unsubstantiated.
- Table 2 shows that iterative rounds improve, but CLIP-Round-4 still underperforms CLIP-longer, which just trains on all data for the same total compute. The paper acknowledges this, but it somewhat undercuts the claim that iterative filtering is the key ingredient. The win comes from combining iterative filtering with continuous training (Self-Filtering), not from the filtered datasets alone.
- The paper does not compare against any Datacomp leaderboard methods beyond the basic CLIP score baseline. Without these comparisons, it is hard to assess where Self-Filtering actually stands.

**Requested Changes:**

Major changes:
- Compare against recent Datacomp baselines. The paper only compares against all data and OpenAI CLIP score filtering. I think the authors should compare against methods like negCLIPLoss (Wang et al., NeurIPS 2024) and T-MARS (Maini et al., ICLR 2024) on the Datacomp medium, to name a few. Without this, the paper cannot credibly claim the method is competitive. The Datacomp medium leaderboard now reaches ~33-40% ImageNet accuracy; the paper's numbers need to be placed in this context.
- Discuss and differentiate from ITLM (Shen & Sanghavi 2019) more thoroughly. The paper cites this work but does not explain how Self-Filtering differs beyond its domain. The core loop (train, score, select top fraction, retrain) is essentially the same. The mixing step and the fact that the model isn't reset are the key differences. These should be stated explicitly, and their contribution should be isolated experimentally (e.g., Self-Filtering without mixing vs. with mixing vs. ITLM-style reset-and-retrain).
- Report compute overhead of the scoring step. Each round requires a full forward pass over all N samples to compute cosine similarities. With R=8 rounds, this adds 8 inference epochs. Report this as a fraction of total training FLOPs. For example, if scoring adds 25-30 % overhead, it would be helpful to know whether the performance gain justifies the added cost compared to simply training longer on all data.
- Address confirmation bias. The paper's single-model self-selection is exactly the setup that co-teaching (Han et al. 2018) was designed to fix. The mixing step partially addresses this, but the paper should: (a) show how the selected set evolves across rounds (what fraction of samples are newly selected vs. retained from the previous round?), (b) measure whether the selected set converges or keeps changing, and (c) discuss whether a two-model variant (analogous to co-teaching) would help.

Changes that would strengthen the paper:
- Ablate the mixing ratio more carefully. The current approach doubles the weight of selected samples (so they appear roughly 2x in the resampled dataset). Try other ratios: 3x, 5x, or soft weighting by cosine similarity score rather than hard top-p selection. The hard threshold at p% is somewhat arbitrary.
- Show what the filtering actually does to the data. Inspect some examples: what gets upweighted, what gets downweighted? Are there systematic patterns? This would build intuition about why the method works and what biases it might introduce.
- The streaming algorithm deserves more attention. It processes non-overlapping chunks sequentially, meaning the filter model only ever sees one chunk ahead. This is quite different from Algorithm 1, which re-scores the entire pool. The relationship between these two variants should be discussed, and ideally, an experiment comparing them on the same data size would be included.

---

> ### Author Response · Authors · 2026-04-04
> **Rebuttal**
>
> ## Discuss ITLM
>
> Our approach is related to ITLM (Shen & Sanghavi 2019) as both iteratively use the training model as a filter. As we note in the related work, we differ in the setting and in the addition of the essential mixing phase. The mixing step is crucial for good performance and avoiding self-confirming biases. Experimentally, Figure 2 shows that the “Self-Filtering” model, which includes the mixing phase, is better than “Filtered Data exclusively”, which trains only on self-selected data (as ITLM).  As the reviewer pointed out, selection models suffer from a confirmation bias, and the mixing step is designed to address it.
>
> ITLM optionally resets the model between selection iterations. In Table 2, we essentially do the same: we reinitialize the training model between iterations to show improvements in data quality across rounds. But resetting wastes training samples. Self-Filtering that continually trains on a new mix achieves much better results (18.9 vs 15.6 average performance) than iterative training with model resets, while seeing the same total number of samples for training the filter and the model.
>
> ITLM uses general small-scale settings like MNIST, CIFAR, FashionMNIST, and CelebA with smaller models like WideResNet-16-10. We use medium-scale internet vision-language datasets using CLIP ViT-B/32 models. While ITLM shows fundamental results of iterative filtering in general image classification and generation settings, we provide evidence for large-scale vision-language settings.
>
> ## Compute overhead
> Please see the “Sample efficiency vs. computational efficiency”  section in the general response above.
>
> ## Confirmation Bias
>
> Self-Training methods that use self-labeling, or self-selecting data, suffer from the effect of collapsing to a small subset, as we discuss in related work paragraphs “Self-Training and collapse” and “Self-Training for filtering”. The mixing phase of Self-Filtering is specifically designed to address this. Bertrand et al.(2024) showed that for generative models, iteratively trained on their own outputs, one necessary condition for not diverging is that the generated data is mixed with real data. This motivates the our mixing stage, which achieves a good balance between exploitation and diversity and avoids collapse.
>
> As pointed out by the reviewer, co-teaching (Han et al. 2018) is designed to explicitly tackle the self-confirming bias by training two models where each one selects data for the other. This relies on the fact that the “two networks have different learning abilities, they can filter different types of error introduced by noisy labels.” Yu et al. (2019) point out that ,over time, the two networks will “converge to a consensus and Co-teaching reduces to the self-training” with similar confirmation-bias problems. For this, they propose Co-teaching+ that applies co-teaching only on samples where the two models disagree. This results nevertheless in low sample efficiency, since most samples are not used. In our case, mixing the real data breaks the confirmation-bias, since the real data will contradict the self-confirming biases. While we have different trade-offs, we believe the mixing strategy has valid motivations for avoiding confirmation bias. Using the mixing stage is beneficial for Self-Filtering in all our experiments.
>
> ## “CLIP-Round-4 still underperforms CLIP-longer,”
>
> In Table 2, each CLIP-Round-i is trained from scratch on a dataset filtered by the previous model. All models improve on each other, proving that iterative filtering yields increasingly better datasets. The reviewer makes a good point that CLIP-longer that trained on 4 times more training iterations, is still better than CLIP-Round-4. Both these models use the same amount of supervision from the same amount of training samples seen, but the resetting of the training model is wasteful in terms of the efficiency of the samples seen. This is the key point in our analysis; we aim for approaches that utilize the training signal effectively and match the number of training examples seen. In all our results, Self-Filtering improves on the unfiltered baselines, at the same number of training examples seen.
>
>
> ## Recent baselines
>
> One key question of the paper is whether the filtering model needs to be stronger than the currently trained model, and thus, whether we need a pre-trained model?
>
> Previous methods don’t tackle this question, as they rely on the knowledge learned during pre-training of the filter, and don’t factor the samples used in the pre-training into the analysis.  We agree that using pre-trained models is more effective, and negCLIPLoss (Wang et al., NeurIPS 2024) and T-MARS (Maini et al., ICLR 2024) are good examples of current methods. negCLIPLoss uses the clip loss as a training score (that includes negative pairs). T-MARS suggests removing text from images to avoid relying on reading it. In the new revision, we’ll cite these papers, noting that they obtain superior performance.

---

### Review · Reviewer_tDVA · 2026-03-20

**Summary Of Contributions:**

Summary:
The paper proposes a bootstrapped data selection mechanism termed "Self-Filtering" for Vision-Language Models (VLMs) like CLIP. To address the heavy reliance on external, pre-trained teacher models (e.g., OpenAI's CLIP) for data curation, the authors suggest an iterative pipeline where a randomly initialized model is trained on a noisy dataset and periodically paused to act as a scoring function (using cosine similarity). The model selects a high-confidence subset of the data, which is then concatenated with the full, noisy dataset and resampled to form the next training batch. This approach aims to balance exploitation (learning from likely-clean data) and exploration (maintaining diversity from the full distribution).

Key Strengths:

Insights into the Exploration-Exploitation Trade-off: The empirical investigation in Section 4.3 provides valuable quantitative backing for industrial data mixing strategies, demonstrating that a blend of high-confidence subsets and the full noisy distribution prevents the model from hitting a generalization ceiling.
Zero-Shot Filtering Transfer: The most compelling contribution (Section 5.1 / Appendix A.2) demonstrates that a lightweight model, evolved through deep bootstrapping on a small dataset, develops a "scoring intuition" that can be transferred to entirely unseen, larger distributions with minimal cost, even outperforming massive external models like OpenAI CLIP ViT-L/14 in specific setups.
Key Weaknesses:

Compute Equivalence Fallacy: The paper claims to use the "same training budget" by matching gradient update steps, completely ignoring the massive hidden computational cost of running full-dataset forward passes during the periodic filtering phases.
The Curriculum Learning Paradox: The narrative heavily relies on the premise of dynamic curriculum learning, which is explicitly contradicted by the authors' own ablation study (Section 5.2), where a static dataset extracted from the final iteration yields better results than the dynamic online process.
Outdated Scoring Mechanism and Baselines: The reliance on raw cosine similarity (CLIPScore) and the lack of comparison against modern, computationally comparable filtering advancements (e.g., negCLIPLoss, DFN, FLYT) severely weakens the paper's claim of being "competitive" in the 2025-2026 research landscape.

**Audience:**

Yes

**Audience Explanation:**

Despite the flaws in the primary narrative and experimental setup, the paper contains highly valuable empirical observations for the TMLR audience, particularly researchers working on Data-Centric AI, VLM pretraining, and self-supervised learning:

The Data Mixing Paradigm: The paper provides a very clear empirical demonstration of why training exclusively on "clean" data leads to representation collapse, and why mixing high-quality data with the noisy, diverse long-tail is strictly necessary for generalization.
Offline Probe Evolution: The finding that a small model, iteratively trained on a tiny subset (12.8M), can be frozen and used as an offline zero-shot filtering probe for a massive, unseen dataset is highly practical. It suggests a new paradigm where researchers can afford to spend heavy compute on a small "sandbox" dataset to evolve a data-scoring probe, which is then cheaply deployed across billion-scale datasets.

**Broader Impact Concerns:**

The paper currently lacks a Broader Impact Statement. Data filtering mechanisms inherently carry the risk of amplifying representational harms. Because the "Self-Filtering" mechanism relies on the model's own evolving biases to select future training data, there is a severe risk of a feedback loop that disproportionately discards minority concepts, non-standard English text, or culturally diverse images (which typically yield lower initial cosine similarity scores). The authors must add a Broader Impact Statement addressing how bootstrapped filtering might exacerbate dataset bias and marginalize underrepresented groups, and discuss potential mitigation strategies (e.g., fairness-aware resampling constraints).

**Claims And Evidence:**

No

**Claims Explanation:**

The core claims of the paper suffer from critical logical paradoxes and unfair empirical setups that undermine the validity of the conclusions:

1. The Compute Equivalence Fallacy:
The authors claim their method is efficient and compares models under the "same training budget" (defined purely by training steps). However, the Self-Filtering mechanism requires the model to periodically halt and perform forward-pass inference over the entire candidate pool (e.g., 128M samples in DataComp Medium) to calculate similarity scores and rank them. For instance, doing this 10 times across a training run incurs an astronomical FLOP and memory I/O overhead. If this massive hidden compute budget were instead allocated to the "All Data" baseline (allowing it to train for several more epochs), the baseline's performance would likely improve significantly. Claiming victory on a skewed computational track invalidates the efficiency claims.

2. The Curriculum Learning Contradiction:
The paper motivates the dynamic, online nature of Self-Filtering as an implicit "curriculum" (Section 5.2). Yet, the authors' own ablation reveals that extracting the final static data mix and training a new model from scratch on this fixed dataset yields a better average performance (19.7%) than the dynamically evolved model (19.2%). This result fundamentally breaks the narrative: if training on a static snapshot is superior, it proves that the online dynamic shifting of data distribution actually harms the optimization landscape (likely due to non-stationarity disrupting momentum-based optimizers).

3. Lack of Contemporary Baselines:
The paper evaluates its method against raw CLIPScore and basic baselines. In the current context of VLM data curation, raw cosine similarity is known to suffer from severe distribution collapse and specificity issues. The paper fails to compare against or integrate contemporary, compute-comparable methods like negCLIPLoss (which normalizes the score using contrastive batch dynamics) or Data Filtering Networks (DFN). Without benchmarking against these modern standards on DataComp Medium, the claim that the method is "effective and competitive" lacks a proper anchor.

4. The Confirmation Bias Loophole:
The authors claim the model learns to distinguish "hard samples" from "noise." However, the filtering criterion is strictly the highest cosine similarity (lowest loss). If a hard sample yields a high loss early in training, it is discarded from the core subset. There is no mathematical mechanism proposed to rescue hard samples later, nor is there a penalty to prevent the model from over-fitting to spurious shortcut features (confirmation bias). The reliance on random global mixing is a heuristic patch, not a principled solution to this well-known self-training flaw.

**Requested Changes:**

To secure a recommendation for acceptance, the following adjustments must be made:

Critical Changes (Required for Acceptance):

Re-frame the Narrative (Abandon Curriculum Learning): The authors must pivot the core narrative away from "dynamic curriculum learning," as it is falsified by their own results in Section 5.2. The paper should be repositioned around the concept of "Offline Probe Evolution for Static Dataset Curation." The value of the method is clearly in finding the optimal static data mixture, not in the online dynamic training process.
Correct the Compute Accounting: The authors must explicitly calculate and report the FLOPs/wall-clock time consumed by the periodic full-dataset inference passes. Furthermore, they must introduce a "Strong Baseline" where the "All Data" model is allowed to train for additional epochs equivalent to the compute time wasted on the Self-Filtering inference steps. The method must demonstrate superiority under true compute equivalence.
Include Contemporary Baselines: The authors must benchmark against modern metric-based filtering methods that do not require massive external datasets, such as negCLIPLoss or s-CLIPLoss. If running new baselines on DataComp Medium is computationally prohibitive, the authors must at least provide a robust theoretical discussion comparing their raw cosine similarity approach to these normalized/gradient-based metrics (e.g., FLYT, DFN), acknowledging the limitations of their current scoring function.
Suggested Changes (To Strengthen the Work):

Address Confirmation Bias: Add a dedicated discussion (or empirical probe) on how the model avoids learning spurious shortcuts during the early phases of Phase 2. How exactly does the model differentiate a true "hard sample" from a "noisy sample" if both exhibit high loss early on?
Clarify the Mixing Mathematics: Provide a more rigorous formalization of the Phase 3 resampling strategy. Connecting this explicitly to the information entropy or gradient variance of the batches would elevate the paper from a heuristic engineering report to a theoretically grounded study.

---

> ### Author Response · Authors · 2026-04-04
> **Rebuttal**
>
> We appreciate and thank the reviewer for their insightful comments and suggestions.
>
> ## Compute Equivalence Fallacy
>
>
> Please see the “Sample efficiency vs. computational efficiency”  section in the general response above.
>
> ## Confirmation Bias.
>
>
> Self-Training methods that use their own prediction to influence their training, by self-labeling, self-generating, or self-selecting data, suffer from a cycle of reinforcement of a small, highly confident subset leading to a collapse effect, as we discuss in paragraphs the “Self-Training and collapse” and “Self-Training for filtering” in the related work. The mixing phase of Self-Filtering is specifically designed to address the confirmation bias. It has been shown that for generative models, iteratively trained on their own outputs, one necessary condition not to diverge is that the mixing of real data with the generated data (Bertrand et al., 2024). This motivates the mixing stage of our method, which achieves a good balance between exploitation and diversity, thus avoiding collapse.
>
>
> In terms of empirical results, all of our experiments show that the mixing stage is beneficial for Self-Filtering across all scales.
>
> >”How exactly does the model differentiate a true "hard sample" from a "noisy sample" if both exhibit high loss early on”
>
> The model cannot distinguish true "hard samples" from "noisy samples" early on; that is why iteratively selecting multiple times while mixing the whole distribution is crucial. Through mixing, the model would still see true “hard samples” later in training, and if at that point it can understand it better, it will learn it in a generalizable way and select it in the future rounds.
>
>
>
>
> ## Self-Filtering doesn’t benefit from the curriculum effects
>
> The reviewer is correct that the curriculum effects don’t show benefits, and, to the contrary, the latest data mix is the best. And this is the exact point we are making. Going through more samples obtains better models, which, in turn, curate better datasets (e.g., “Section 4.2 Investigation: Iteratively selected datasets yield progressively better models”). Then, in “Section 5.2 Benefits of Self-Filtering are not due to curriculum learning”, we further explain that the final data mix is the best; if we train on it from the beginning, we achieve better results. But this uses twice the amount of samples, once for training the filter and once for training the final model. Thus, Self-Filtering *does not benefit from a curriculum effect*, but when taking into account the samples seen by the filtering method, it is more seen sample efficient to apply Self-Filtering.
>
>
> ## Contemporary Baselines
>
> We agree that we don’t compare to strong baselines. We note that most SOTA results don’t account for the number of samples used for filtering, and our purpose is to find if filtering can obtain improvements when accounting for the number of samples used for filtering. We don’t claim that our method obtains SOTA or competitive results with current Datacomp leaders; we will make this clearer in the revision.

---

### Author Response · Authors · 2026-04-04
**Rebuttal**

# General Response

We thank all reviewers for their careful reading of our submission and for providing on-point feedback. We address their comments separately below, but first, we provide some common observations.

## Framing and contribution

We start by clarifying the work's framing and contribution.

The paper focuses on one central question: what happens when we do not have access to a pre-trained model to filter a vision-language dataset? Regardless of the exact method, all current methods rely on an existing pre-trained model for data selection. We don’t claim the method achieves competitive results, nor do we make claims about being *computationally* efficient. We simply investigate whether a filtering method needs extra knowledge from a pre-trained model to be effective, or whether the training model itself, with no such advantage, can serve as an effective filter. We show the latter: the training model is effective in filtering.

The contributions of the paper include:

- We show that effective filtering strategies can be achieved without using pre-trained models or additional data.

- We propose a data selection method consisting of iteratively training and improving the training dataset, and show that it is effective: it produces better filters, inducing better datasets that, in turn, produce *self-improving models*.

- We refine the prior findings that, in general, filtering and downstream performance are not correlated (Fang et al. 2023). Consistent with this finding, we show that the data obtained by Self-Filtering using a smaller and less capable model is sometimes as good as data filtered by the bigger and more capable pre-trained model (Table 1, Figure 4, Figure 5). Nevertheless, across a training run, the intermediate checkpoints have their filtering and downstream performance correlated.

- We suggest a strategy of mixing selected samples with random examples, balancing the exploitation of learned knowledge with the diversity available in the whole dataset. This addresses the known self-confirmation collapse of self-training. This is still a very relevant problem in the context of large models trained on data filtered by previous versions of the model.

---

> ### Author Response · Authors · 2026-04-04
> **Comment on efficiency**
>
> ## Sample efficiency vs. computational efficiency
>
> What makes a filtering method based on a fixed model effective? Is it the fact that the filtering model has more knowledge than the training model? Or does the filtering method better exploit the structure of the data, with minimal additional information? To find this, we explicitly take into account the number of training samples seen for the filtering model. Given a baseline trained on a fixed number of samples seen, a filtering method that uses the same number of samples and produces better results is said to be more seen sample-efficient. This is the main goal of the paper, and for this, we propose a Self-Filtering approach.
>
>
> To be clearer, let us establish some terminology. We distinguish between:
>
> A method is more **sample efficient (with unique samples)** if it reaches better performance than another method using the same number of unique samples.
>
> We denote as **seen sample efficient** if it reaches better performance than another method using the same number of total samples seen, taking repetitions into account. This can also be viewed as **optimization efficiency**.
>
> A method is more **computationally efficient** if it reaches better performance than another method using the same amount of computation.
>
> In the most common large-scale scenario, where the data is used only once (as in our streaming setting), the first two notions coincide.
>
> The first two notions are fundamentally important, and only once a model is sample efficient does it make sense to investigate its computational efficiency. In this work, we evaluate only **seen sample efficiency**.
>
> ### *Why this question is non-trivial*
>
> It is not obvious that a self-filtering approach, without additional information from a pre-trained model, knowledge of the noise level, or knowledge of the clean-data distribution, can be more sample efficient than no filtering at all. To our knowledge, there are no other works addressing this question in the context of vision-language training; our work is intended to provide some initial answers.
>
> ### *Limitation regarding compute*
>
> As already noted in the Limitations section of the original manuscript, we do not account for the computational cost of scoring, and therefore, we cannot make claims about computational efficiency. We will make this point clearer in the revision.
>
> ### *Practical considerations*
>
> On the practical side, we note that methods in the DataComp benchmark also do not account for the computational time required for scoring samples. From a practical perspective, this can be justified by the fact that scoring can often be performed in parallel on older devices that are not being used for training. Since inference requires less memory and is faster than training, it can be run on older hardware, which is often either more available in most institutions or significantly cheaper. For example, an H100 SXM with 80GB VRAM costs about \\$2.69/hr, whereas an RTX A6000 with 48GB VRAM costs about \\$0.33/hr [A].
>
> Since Self-Filtering requires scoring only at fixed intervals, for both multi-epoch and streaming settings, this scoring can be carried out in parallel with limited communication overhead.
>
>
>
> [A] “GPU Pricing.” RunPod, https://www.runpod.io/gpu-pricing. Accessed 3 Apr. 2026
>
> ## Broader Impact Statement
>
> We also thank the reviewers for their valuable suggestions on the broader impact. We agree that vision-language models and their curation warrant a broader impact assessment, and introduced a statement in the revision.

---

### Decision · Action_Editor_LWtN · 2026-05-25

**Recommendation:** Accept as is

**Audience:**

Yes

**Audience Explanation:**

All reviewers agreed that this is a timely and interesting topic for TMLR readers working on VLMs.

**Claims And Evidence:**

Yes

**Claims Explanation:**

The paper investigates a self-filtering strategy for VLMs. While the reviewers found the claims in the orginal submission to be exaggerated, the authors' rebuttal has addressed those concerns. The clarified and sharpened claims that the authors made in the rebuttal seem to be well-supported by the presented evidence. One reviewer (tDVA) was unfortunately unresponsive after the rebuttal, but it is the AE's belief that their concerns have also been addressed. We would still recommend that the authors take the reviewer feedback into account when preparing the camera-ready version.